Evaluation of the retreatability of bioceramic root canal sealers with various formulations in simulated grooves

Sümbüllü Meltem meltem_endo@hotmail.com 1
Ali Afzal 2
Bakhsh Abdulaziz 3
Arslan Hakan 4
1 Faculty of Dentistry, Atatürk University , Erzurum , Türkiye
2 Pacific Dental College and Hospital , Udaipur , India
3 Faculty of Dental Medicine, Umm Al-Qura University , Makkah , Saudi Arabia
4 İstanbul Medeniyet University , İstanbul , Türkiye
Abu Hasna Amjad
Electronic publication date: 2025 Dec 2
Publication date: 2025
Volume: 13
Electronic Location ID: e20398
Received 2025 Aug 19; Accepted 2025 Oct 27
Copyright: ©2025 Sümbüllü et al.
Copyright year: 2025
Copyright holder: Sümbüllü et al.
License: This is an open access article distributed under the terms of the Creative Commons Attribution License, which permits unrestricted use, distribution, reproduction and adaptation in any medium and for any purpose provided that it is properly attributed. For attribution, the original author(s), title, publication source (PeerJ) and either DOI or URL of the article must be cited.
License URL: https://creativecommons.org/licenses/by/4.0/

Keywords: Calcium silicate based sealer, Passive ultrasonic irrigation, Retreatability

Funding: The authors received no funding for this work.

==============================
Background

Bioceramic sealers are increasingly used due to their favorable properties, but their strong bonding to dentin complicates retreatment procedure. In addition, canal irregularities such as isthmuses make the complete removal of filling materials more challenging.

Aim

This study aimed to evaluate the retreatability of five bioceramic sealers (BioRoot RCS, Bio-C Sealer, CeraSeal, Endoseal MTA, and AH Plus Bioceramic Sealer) using passive ultrasonic irrigation.

Materials and Methods

Sixty human mandibular premolars with straight canals were prepared to size 40/0.04 taper. After longitudinal splitting, standardized grooves were created. The root canals were subsequently obturated using the single cone technique and stored at 37 °C with 100% humidity for 21 days. Passive ultrasonic irrigation was used to assess sealer removal. Statistical analysis was performed with Kruskal–Wallis and Mann–Whitney U tests.

Results

The amount of residual sealer in the apical region was statistically higher compared with the coronal region. In the coronal region, the highest amount of residual material was observed in the Bio-C Sealer group, and this difference was statistically significant compared to the other groups. There was no significant difference among the other groups. In the apical region, there were no statistically significant difference between Bio-C Sealer and Endoseal MTA; however Bio-C Sealer showed a statistically significantly higher amount of residual material compared to BioRoot RCS, Ceraseal, and Ah Plus Bioceramic Sealer.

Conclusion

Among the evaluated bioceramic sealers, Bio-C Sealer exhibited the highest amount of residual material, particularly in the apical region, indicating its lower retreatability. Passive ultrasonic irrigation improved the removal of BioRoot RCS, CeraSeal, and AH Plus Bioceramic Sealer, but was less effective for Bio-C Sealer and EndoSeal MTA. These findings highlight the variability in retreatability among different bioceramic sealers and emphasize the importance of material selection in cases with potential need for retreatment.

Introduction

The prevalence of apical periodontitis in root filled teeth is as high as 39% (Tibúrcio-Machado et al., 2021). The most common causes of treatment failure include inadequate canal shaping, insufficient disinfection, and poor obturation quality (Gorni & Gagliani, 2004). In such cases, the preferred approach is non-surgical endodontic retreatment (Torabinejad et al., 2009). Several factors influence the success of retreatment, including the complex anatomy of the root canal system, the condition of the periapical tissues, and the type of root canal filling material previously used (Wong, 2004). In recent years, bioceramic-based root canal sealers have gained popularity, especially when used with the single-cone obturation technique (Collado-Castellanos et al., 2025). In this method, the volume of filling material inside the canal is higher than in other techniques, which may effect the retreatment process (Eymirli et al., 2019).

There are various formulations, preparation methods, application techniques, and chemical compositions of bioceramic materials. CeraSeal (Meta Biomed Co., Cheongju, Korea) is a pre-mixed bioceramic sealer that offers favorable biocompatibility, exhibits high filling capability, and demonstrates superior physicochemical properties and homogeneity compared to powder–liquid CS sealers (Kharouf et al., 2020). Bio-C Sealer (Angelus, Londrina, PR, Brazil) is a newly developed, premixed, ready-to-use root canal sealer primarily composed of calcium silicates. Bio-C Sealer exhibits superior penetration into dentinal tubules and improved adaptation due to its smaller particle size, enhanced flowability, and hydrophilic properties (Bademela et al., 2024). BioRoot RCS (Septodont, Saint- Mouer-Dis-Fosses, France) is a calcium silicate-based sealer with antimicrobial properties due to the release of calcium hydroxide (Siboni et al., 2017). Published research shows that this material provides successful results in terms of biocompatibility and bioactivity (Camps et al., 2015). EndoSeal MTA (Maruchi, Wonju, Korea) is a ready-to-use composed of calcium silicate, calcium aluminate, calcium sulfate, and a radiopaque agent (Adl, Shojaee & Pourhatami, 2019). It has shown excellent physical characteristics, biocompatibility, strong adhesive properties, minimal discoloration, and superior distribution as a sealer (Kim et al., 2018). AH Plus Bioceramic Sealer (Dentsply Sirona, Charlotte, NC) is a ready-to-use calcium silicate–based root canal sealer. As stated by the manufacturer, it offers rapid setting time, strong resistance to washout, and good radiopacity, while also being safe, biocompatible, and non-discoloring to the tooth structure (Souza et al., 2023).

Variations in the chemical composition of calcium silicate–based materials contribute to the complexity of retreatment procedures (Donnermeyer et al., 2018). Particle size, flow characteristics, and solubility directly affect the removal of sealers from the root canal system (Zordan-Bronzel et al., 2019; Bademela et al., 2024). The choice of vehicle—such as water, saline, or propylene glycol—may further influence the characteristic properties of bioceramic sealers (Holland et al., 2007). In addition, the proportions of cement, radiopacifying agents, and other functional components play a decisive role in determining their chemical behavior (Cardinali & Camilleri, 2023). These differences highlight the clinical importance of understanding the retreatability of newly introduced bioceramic sealers. For instance, the high zirconium oxide content of Ceraseal and the inclusion of dimethyl sulfoxide (DMSO) in AH Plus Bioceramic Sealer may facilitate their removal, whereas the chemical bonding tendency of EndoSeal MTA and the smaller particle size with enhanced flowability of Bio-C Sealer may complicate the retreatment process (Neelakantan, Grotra & Sharma, 2013; Ballal et al., 2019; Cardinali & Camilleri, 2023). To the best of our knowledge, there are no previous studies that have investigated the removal of five different bioceramic sealers from root canals. Therefore, this study evaluates the retreatability of five different bioceramic endodontic sealers (Bio Root RCS, Bio-C Sealer, Ceraseal, Endoseal MTA and AH Plus Bioceramic Sealer) from artificially created grooves in mandibular premolar teeth using passive ultrasonic irrigation activation. The null hypothesis of this study was that there would be no significant differences among the five tested bioceramic sealers regarding their retreatability from artificially created grooves, especially anatomic irregularities.

Materials & Methods

This manuscript was written according to the Preferred Reporting Items for Laboratory Studies in Endotology (PRILE) 2021 guidelines. The study protocol was reviewed and approved by the Institutional Review Board of the Faculty of Dentistry, Atatürk University (decision No. 82-2024/12). The recruitment period started on January 1, 2025, and was completed on February 29, 2025. All teeth used in this study were extracted due to periodontal indications, in accordance with the Declaration of Helsinki.

Seventy-five extracted human single-rooted teeth with intact apices, a minimum root length of 20 mm, and a single oval-shaped canal were selected. The sample size was determined using G∗Power 3.1.9.4 software (Düsseldorf University, Düsseldorf, Germany) with an alpha error probability of 0.05, an effect size of 0.6 and a power of 0.95 (Gouveia et al., 2025). A sample size of 12 specimens per group was calculated; however, to account for a potential 25% drop-out rate, this number was increased to 15 specimens per group, resulting in a total of 75 specimens. Buccal and proximal radiographs were taken to confirm the presence of a single canal. Teeth with curved canals, calcifications, previous endodontic treatment, or wide canals (initial apical size greater than #30) were excluded from the study. The working length (WL) was standardized to 18 mm. Canal shaping was performed by an experienced endodontist using the NeoNiti A1 rotary file system (Neolix, Châtres-la-Forêt, France) up to size 40.04 at the WL at 500 rpm and 1.5 Ncm. A size 10 K-file (Dentsply; Maillefer, Switzerland) was used to maintain apical patency during the procedure. Throughout instrumentation, canals were irrigated with two ml of 3% sodium hypochlorite (NaOCl) using a close ended side-vented needle (Endo-Top, Cerkamed, Stalowa Wola, Poland).

Following canal instrumentation, each root canal was irrigated with five ml of 3% NaOCl (Imicryl, Konya, Türkiye), then flushed with five ml of 17% ethylenediaminetetraacetic acid (EDTA) (Imicryl). The specimens were stabilized in Eppendorf tubes using impression material. Once removed from the silicone base, the roots were sectioned longitudinally with a diamond disc to expose the mesial and distal canal walls, taking care to avoid entering the root canal space. Two standardized artificial grooves, measuring four mm in length, 0.2 mm in width, and 0.5 mm in depth, were created on opposite canal walls using a Cavitron tip (Satelec, Acteon Group) under a stereomicroscope (Carl Zeiss Meditec AG) to simulate uninstrumented canal irregularities (Lee, Wu & Wesselink, 2004; Grischke, Müller-Heine & Hülsmann, 2014; Ballal et al., 2019). These grooves were prepared at two different levels: one group was located between 2 and 6 millimeters from the apex, representing the apical region, while the other group was located between 10 and 14 millimeters from the apex, representing the coronal region. To eliminate the smear layer from the groove surfaces, an additional irrigation with two ml of 17% EDTA was performed. The root segments were then reassembled, with the apical foramen and outer surfaces sealed with wax, and placed into Eppendorf tubes.

The samples were randomly divided into five groups according to sealer type (n = 15): CeraSeal, Bio-C Sealer, BioRoot RC Sealer, Endoseal MTA, Ah Plus Bioceramic Sealer. Final irrigation was performed with 5 mL of 17% EDTA, followed by distilled water. The root canal drying protocol was performed using Capillary tips in white, purple, and green (Ultradent Products Inc., South Jordan, UT, EUA), respectively, along the entire length of the root canal, to maintain the residual moisture necessary to enhance the adhesion of bioceramic sealers to the intraradicular dentin, as demonstrated in previous studies (Ozlek et al., 2020; Pelozo et al., 2023). At the end of the drying protocol, the root canals were filled using the single-cone technique. The access cavity was sealed with Cavit (3M ESPE, Seefeld, Germany). The specimens were then kept in Hank’s balanced salt solution at 37 ∘C for a period of three weeks to allow complete setting of the sealer.

After a 21-day incubation period, the temporary restoration was removed, and the retreatment procedure was performed using  the HyFlex Remover instrument (#30.07) at 800 rpm and two Ncm, followed by HyFlex CM instruments (#40.04 and #50.04) (Coltene, Altstätten, Switzerland) at the WL, powered at 400 rpm and two Ncm. Irrigation with 3% NaOCl was performed between the use of each file. Final irrigation was activated with a size of #20 Irrisafe tip (Satelec, Acteon Group) attached to a Suprasson P5 Booster ultrasonic unit (Satelec, Acteon Group, Merignac, France) set at power level 4, following the manufacturer’s guidelines. The irrigant was introduced into the canal, with the tip placed one mm short of the WL, and activated using a five mm vertical oscillating motion for 30 s. This procedure was repeated using three ml of 3% NaOCl twice for 30-second, followed by three ml of 17% EDTA applied in the same manner, resulting in total 12 ml of irrigant over 2 min.

The root canals were dried with paper points, and the roots were dissembled to evaluate the amount of root canal sealer in the artificial grooves. The amount of root canal sealer in each artificial groove was evaluated using a microscope with 24 × magnification (Carl Zeiss Meditec AG, Jena, Germany). The clinician performing the procedure was blinded to the type of sealer present in each sample. The residual sealer on the groove was evaluated and scored by two independent evaluators, according to the scoring system proposed by Ballal et al. (2019) (Fig. 1): sealer distribution was scored on a scale from 0 to 3, where 0 indicated no sealer in the groove, one indicated sealer in less than half of the groove, two indicated sealer in more than half, and three indicated complete filling of the groove.

Figure 1 Examples of the different score scales.

Sealer distribution was scored on a scale from 0 to 3, where 0 (Panel A) represented the absence of sealer within the groove, 1 (Panel B) represented sealer present in less than half of the groove, 2 (Panel C) represented sealer present in more than half of the groove, and 3 (Panel D) represented complete filling of the groove.

Statistical analysis

Residual sealer was scored by two endodontists who were unaware of each other. This process was repeated one week later. The Cohen kappa value was tested for inter/intra-observer agreement. The scored data were analysed with the Kruskal Wallis test and Mann whitney-U with the level of significance set at the 95% confidence interval (p = 0.05).

Results

The PRILE 2021 flowchart is presented in Fig. 2. The intra-individual reproducibility for the first and second observers was 95% and 97%, respectively. The agreement between the two researchers showed high reliability for the apical and coronal regions, with Cohen kappa values of 0.97 and 0.98, respectively.

Figure 2 The PRILE 2021 flowchart.

Figure 3 presents the residual sealer scores for all groups. The amount of residual sealer in the apical region was higher compared to the coronal region (p = 0.032). In the coronal region, the highest amount of residual material was observed in the Bio-C Sealer group (p < 0, 05), and this difference was statistically significant compared to the other groups. There was no significant difference among the other groups (p > 0, 05) (Table 1).

Figure 3 The residual sealer scores for all groups in the canals.

Table 1 Results of the Mann–Whitney U tests between groups at coronal region.

In the coronal region, the highest amount of residual material was observed in the Bio-C Sealer group, and this difference was statistically significant compared to the other groups. There was no significant difference among the other groups.

Groups			CeraSeal	Bio-C Sealer	BioRoot RCS	Endoseal MTA	Ah Plus Bioceramic Sealer	
CeraSeal	–	<0.001	0.16	0.977	0.932	
Bio-C Sealer	<0.001	–	<0.001	<0.001	<0.001	
BioRoot RCS	0.16	<0.001	–	0.242	0.089	
Endoseal MTA	0.977	<0.001	0.242	–	0.887	
Ah Plus Bioceramic Sealer	0.932	<0.001	0.089	0.887	–	
Notes.

The bold text indicates statistically significant differences between the groups.

In the apical region, there were no statistically significant difference between Bio-C Sealer and Endoseal MTA (p < 0, 05); however Bio-C Sealer showed a statistically significantly higher amount of residual material compared to BioRoot RCS, Ceraseal, and Ah Plus Bioceramic Sealer (p > 0, 05). No statistically significant differences were found between Endoseal MTA and other groups (p > 0, 05) (Table 2).

Table 2 Results of the Mann–Whitney U tests between groups at apical region.

In the apical region, there were no statistically significant difference between Bio-C Sealer and Endoseal MTA; however Bio-C Sealer showed a statistically significantly higher amount of residual material compared to BioRoot RCS, Ceraseal, and Ah Plus Bioceramic Sealer.

Groups			CeraSeal	Bio-C Sealer	BioRoot RCS	Endoseal MTA	Ah Plus Bioceramic Sealer	
CeraSeal	–	0.033	0.178	0.799	0.143	
Bio-C Sealer	0.033	–	0.003	0.114	<0.001	
BioRoot RCS	0.178	0.003	–	0.143	0.590	
Endoseal MTA	0.799	0.114	0.143	–	0.143	
Ah Plus Bioceramic Sealer	0.143	<0.001	0.59	0.143	–	
Notes.

The bold text indicates statistically significant differences between the groups.

Discussion

Bioceramic-based sealers are notable for their biocompatibility and antimicrobial properties; however, their strong chemical bonding to dentin presents challenges during removal (Arul et al., 2022). This study evaluated the effectiveness of the removal of five different calcium silicate based sealers from artificially created grooves in oval-shaped canals using passive ultrasonic irrigation. Our results showed that the highest amount of residual sealer in both the coronal and apical regions was observed with Bio-C Sealer. In the apical region, no significant difference was found between Bio-C Sealer and EndoSeal MTA. Therefore, the null hypothesis was rejected.

According to the findings of the present study, Bio-C Sealer left a greater amount of residual material in both the coronal and apical regions compared to other calcium silicate–based sealers. Bio-C Sealer is a novel, premixed, ready-to-use sealer primarily composed of calcium silicates. A previous study has reported that, due to its smaller particle size, enhanced flowability, and hydrophilicity, Bio-C Sealer exhibits greater penetration and improved adaptation to dentinal tubules compared to AH Plus sealer (Bademela et al., 2024). Root canal sealers should exhibit flow values that meet the requirements of the ISO 6876:2012 standard (>17 mm). In a previous study, a similar flow value was reported for AH Plus (approximately 20 mm), whereas a higher flow value was observed for Bio-C Sealer (approximately 31 mm) (Zordan-Bronzel et al., 2019). Bio-C Sealer also contains propylene glycol as a carrier (Cardinali & Camilleri, 2023). Propylene glycol is a viscous carrier, and it is known that viscous carriers are less soluble compared to aqueous carriers (Fava & Saunders, 1999). Furthermore, calcium aluminate may be present in low amounts (<5%) in Bio-C sealer, similar to that found in industrial Portland cement (Cardinali & Camilleri, 2023). These compounds exhibit hydraulic properties, and it has been suggested that their presence may enhance the acid resistance and reduce the solubility of tricalcium silicate-based systems (Primus et al., 2021). It is possible that the composition of Bio-C Sealer, along with its micromechanical and chemical bonding properties, contributed to the greater amount of residual material left behind.

Lee et al. (2022) investigated the removal of EndoSeal MTA and found a significantly higher amount of residual material compared to AH Plus. Another study reported that EndoSeal MTA exhibited the highest residue levels particularly in the apical third of the canal (Kim et al., 2019). This characteristic likely linked to its interaction with dentin. Upon contact with dentinal fluid, MTA can induce the formation of carbonated apatite or hydroxyapatite, leading to chemical bonding with the dentinal wall (Neelakantan, Grotra & Sharma, 2013). The results of the current study align with these earlier findings and suggest that such physicochemical interactions may contribute to the challenges associated with removing EndoSeal MTA from the root canal system.

There were no statistically significant differences found between BioRoot RCS, Ceraseal and Ah Plus Bioceramic sealer in the coronal and apical regions. Baranwal et al. (2021) showed that no statistically significant difference was found in the amount of residual sealer between AH Plus and BioRoot RCS after retreatment at various root canal levels. Marchi, Scheire & Simon (2020) showed that removal of the root filling material was successful in 91.67% of the canals treated with BioRoot RCS. BioRoot RCS has a shorter setting time and lower flow (Viapiana et al., 2016), which may have influenced its adaptation to dentin and its retrievability.

The sealers incorporated various radiopacifying agents, including zirconium oxide, tantalum oxide, ytterbium trifluoride, bismuth oxide, and calcium tungstate, often in concentrations exceeding 50%. The high proportion of these fillers inevitably reduces the relative content of cement and other functional additives within the sealer composition. Ceraseal contains %50 of zirconium oxide and it can be contributed easily removal from dentinal surface (Cardinali & Camilleri, 2023). Ah Plus Bioceramic sealer contains only 5–15% tricalcium silicate, in contrast to the conventional calcium silcate based sealer, which typically contains 7–15% dicalcium silicate and 20–35% tricalcium silicate (Cardinali & Camilleri, 2023). Furthermore, AH Plus Bioceramic sealer contains 10–30% dimethyl sulfoxide (DMSO) as a carrier, which may contribute to its easier removal from the root canal (Cardinali & Camilleri, 2023). While concentrations of 4% or higher DMSO impaired bond strength over time, the resin containing 2% DMSO demonstrated higher dentin bond strength (Stape et al., 2016). Shim et al. (2025) showed that Ah Plus Bioceramic Sealer and Ceraseal demonstrated higher mean percentages of removed filling volume at 94.8%, and 92.5%, respectively, compared to 87.1% for the Ah Plus sealer. The obtained findings are consistent with previous studies. The high concentration of zirconium oxide in Ceraseal, along with the presence of DMSO and the low proportion of tricalcium silicate in AH Plus Bioceramic sealer, may have influenced their interactions with dentin, thereby affecting their retreatability.

Passive ultrasonic irrigation (PUI) is a supplementary procedure developed to clean the root canal with an irrigation wave through acoustic flow and cavitation (Van der Sluis et al., 2007). The simultaneous use of PUI and cleaning solutions enhances the flushing effect of the irrigants and increases their flow (Van der Sluis et al., 2007). The positive effect of PUI can be attributed to its vibration, which results in continuous movement of the irrigant within the canal and facilitates the removal of filling materials.

A greater amount of residual sealer was observed in the apical region compared with the coronal region. Because bioceramic sealers interact with calcium in dentin, the use of EDTA may facilitate their separation from the canal walls (Garrib & Camilleri, 2020). Furthermore, the irrigant is more efficient in the coronal portion due to enhanced fluid dynamics and less anatomical complexity, which promotes better dissolution and dislodgement of filling materials (Dioguardi et al., 2018). Accordingly, the greater removal of sealer observed in the coronal region in the present study may be attributed to the enhanced mechanical and chemical effects of EDTA.

Studies evaluating the retreatability of root canal filling materials used different observation methods such as scanning electron microscopy (SEM), light microscope, digital camera, micro-computed tomography (micro-CT), or cone-beam computed tomography (CBCT) (Fenoul, Meless & Pérez, 2010; Neelakantan, Grotra & Sharma, 2013; Grischke, Müller-Heine & Hülsmann, 2014; Suk et al., 2017; Donnermeyer et al., 2018). In the present study, following the longitudinal splitting of the samples, residual sealer on the grooves were assessed using stereomicroscope images through direct visual scoring. Use of digital camera and light microscope can be adequate for the evaluation of cleanliness of the simulated grooves as they are standardized areas of low volume (Ballal et al., 2019). This approach does not require any specialized equipment (Somma et al., 2008). Additionally, this method provided the benefit of enabling reproducibility between different observers. Further studies using novel research techniques such as micro-CT are required to evaluate the retreatability of tricalcium silicate based sealers from root canals.

In this study, all possible variables were kept consistent. Achieving standardization in irrigation volumes and timing remains a key challenge in studies of this nature. However, the total volume and application intervals of the irrigants were kept consistent across all groups. The limitation of the study is the exclusive use of straight root canals, which may restrict the applicability of the findings to curved canals. Another limitation of our study is that, to ensure standardization, all canals were uniformly enlarged to size 40.04 without considering the initial file size. Future studies should investigate the retreatability of bioceramic sealers in curved root canals or in teeth with more complex anatomies, such as isthmuses and oval canals.

Conclusions

Complete removal of the root canal sealers from the grooves could not be achieved. Passive ultrasonic irrigation demonstrated superior retreatment efficacy in the apical region for BioRoot RCS, Ceraseal, and AH Plus Bioceramic Sealer when compared to Bio-C Sealer and Endoseal MTA. In the coronal region, the removal of Bio-C Sealer was found to be less effective.

Supplemental Information

Supplemental Information 1 Dataset

The authors declare that they have no conficts of interest.

Additional Information and Declarations

Competing Interests

Author Contributions

Human Ethics

Data Availability

The authors declare there are no competing interests.

Meltem Sümbüllü conceived and designed the experiments, prepared figures and/or tables, authored or reviewed drafts of the article, and approved the final draft.

Afzal Ali performed the experiments, authored or reviewed drafts of the article, and approved the final draft.

Abdulaziz Bakhsh performed the experiments, authored or reviewed drafts of the article, and approved the final draft.

Hakan Arslan conceived and designed the experiments, analyzed the data, authored or reviewed drafts of the article, and approved the final draft.

The following information was supplied relating to ethical approvals (i.e., approving body and any reference numbers):

Ethics Committee of Faculty of Dentistry, Ataturk University granted Ethical approval to carry out the study within its facilities (Ethical Application Ref: 82-2024/12).

The following information was supplied regarding data availability:

The raw measurements are available in the Supplementary File.

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
