# Peer review of "Evaluation of the retreatability of bioceramic root canal sealers with various formulations in simulated grooves"

_PeerJ, doi:10.7717/peerj.20398_

## Round 0.1 · original submission · Major Revisions

· Academic Editor

Major Revisions

This manuscript addresses an important and controversial issue regarding sealer remnants and their impact on endodontic treatment outcomes. However, major revisions are required. The introduction lacks adequate references and clear rationale, and methodology is insufficiently detailed. Sample size calculation, instrument selection, canal preparation protocol, irrigation strategy, and drying technique must be clarified. The retreatment procedure and measurement method for sealer remnants are outdated, and more reliable approaches (e.g., micro-CT or CLSM) are recommended.

Reviewer 1 ·

Basic reporting

This article is well-written and discusses a fundamental and controversial topic in the field of the impact of sealer remnants on endodontic treatment failure.

Some paragraphs in the introduction lack references; they should be added.

The Background section in the abstract should be shortened.

References for the methodology were not provided.

Experimental design

All the root canal sealers used are commercial forms of calcium silicate-based root canal sealer, not distinct types of these materials. It is necessary to specify the names of the manufacturing companies.

How was the sample size calculated—was it done using G*Power software? Additionally, what was the reference study used for sample size calculation and for deriving the required effect size?

In the section pertaining to root canal preparation, the canals were prepared incorrectly. It is not feasible to prepare all teeth to a single size of 40/04. Instead, each tooth should be prepared to a size that corresponds to its initial apical measurement.

Why was the final irrigation not performed after re-joining the tooth halves and before obturation? Sectioning the tooth longitudinally and bonding it could potentially affect the adaptation of the sealer. Furthermore, the method of canal drying was not mentioned. Given that calcium silicate-based sealers are moisture-friendly, how was an appropriate environment prepared for these materials?

During the retreatment procedure, was a dedicated rotary file system used for retreatment preparation, or was the last instrument used in the initial preparation employed instead?

Validity of the findings

The method used to measure the sealer remnants within the artificial grooves is outdated and no longer currently employed. Furthermore, the reference study cited (Lee, Wu and Wesselink, 2004) when describing this method utilized it to measure dental debris resulting from preparation within the root canal using high-magnification microscopy. It was not used to measure sealer remnants within artificial grooves.

Was magnification used during the measurement of the sealer remnants?

It would be preferable to reevaluate the results using modern and more reliable methods, such as Micro-CT, Confocal Laser Scanning Microscopy (CLSM).

·

Basic reporting

Introduction:
What is the rationale or background of your study? In this introduction it is not clear why this research was conducted.

Experimental design

Methods:
In this study, the authors retreatment
procedure was performed using a size 40.04 file.

1. Why the authors using this method since many studies have reported the efficacy, cleaning ability, and safety during endodontic filling removal using rotary nickel–titanium retreatment instruments.
There are many techniques and materials commonly used to remove sealers.

2. Why the residual sealer on the canal walls were assessed using stereomicroscope images through direct visual scoring? How to distinguish between residual sealer and debris?

3. Many studies found that only SEM permits comprehensive observation of root canal filling remnants or debris.
Another study used CBCT to determine residual sealer.

4. In this study, the authors observed residual sealer in coronal and apical region, but no explanation was written in METHODS, furthermore why residual sealer in coronal and apical region were not compared?

5. Residual sealer was scored by two endodontists who were unaware of each other. This process
was repeated one week later.

Why the scored process was repeated??

Validity of the findings

-

Additional comments

-

---

## Round 0.2 · accepted · Accept

· Academic Editor

Accept

Congratulations! The authors have thoroughly addressed all reviewer comments and substantially enhanced the manuscript. The revision demonstrates clear and concise reporting, with well-organized sections, coherent flow, and appropriately updated references.

·

Basic reporting

The revision already clear

Experimental design

The revision already clear

Validity of the findings

The revision already clear

Additional comments

-